# A First-line management team's strategies for sustaining resilience in a specialised intensive care unit—a qualitative observational study

Karl Hybinette [1,2] Karin Pukk Härenstam,[1,3] Mirjam Ekstedt [1,4]

¹Department of Learning Informatics Management and Ethic, Karolinska Institute, Stockholm, Sweden
²Astrid Lindgrens Childrens Hospital, Karolinska Hospital, Stockholm, Sweden
³Paediatric Emergency Department, Karolinska University Hospital, Stockholm, Sweden
⁴Linnaeus University, Kalmar, School of Health and Caring Sciences, Sweden

**Correspondence to**
Karl Hybinette;
karl.hybinette@ki.se

## ABSTRACT

**Objectives** Acute care units manage high risk patients at the edge of scientifically established treatments and organisational constraints while aiming to balance reliability to standards with the needs of situational adaptation (resilience). First-line managers are central in coordinating clinical care. Any systemic brittleness will be evident only in retrospect through, for example, care quality measures and accident statistics. This challenges us to understand what successful managerial strategies for adaptation are and how they could be improved. The managerial work of balancing reliability and adaptation is only partially understood. This study aims to explore and describe how system resilience is enhanced by naturally occurring coordination performed in situ by a management team under variable circumstances.

**Design** An explorative observational study of a tertiary neonatal intensive care unit (NICU) in Sweden. One year of broad preparatory work followed by focused shadowing observations of coordination analysed through inductive–deductive content analysis from a perspective of resilience engineering.

**Participants** A team of managers (ie, clinical coordinators, head nurses, senior medical doctors).

**Results** The results describe a functional relationship between operational stress and a progression of adjustments in the actual situation, expressed through recurring patterns of adaptation. Managers focused on maintaining coherence in escalating problematic situations by facilitating teamwork through goalsetting, problem-solving and circumventing the technical systems' limitations.

**Conclusions** Coordination supports a coherent goal setting by increased team collaboration and is supported by team members' abilities to predict the behaviour of each other. Our findings suggest that in design of future research or training for coordination, the focus of assessment and reflection on adaptive managerial responses may lie on situations where the system was 'stretched' or 'needed reorganisation' and that learning should be about whether the actions were able to achieve short-term goals while preserving the long-term goals.

## BACKGROUND

Maintaining quality and safety is an ongoing challenge for hospital managers whose units are tasked with delivering increasingly specialised care under complex conditions, such as simultaneously managing acute admissions, staff shortages and rapidly deteriorating patients with life-threatening conditions.[1 2] The neonatal intensive care unit (NICU) is a case in point, as it is highly specialised and serves as its own emergency department, intensive care unit and pre- and postoperative ward, requiring a wide array of interrelated multi-professional teams to operate in a coordinated fashion.

Safety is often defined as 'the absence of preventable harm to a patient during the process of healthcare'.[3] During the last decades of patient safety research, a movement has emerged towards an inclusive view of supporting the healthcare systems' ability to sustain its required operations, the resilience engineering (RE) perspective.[4]

A tenet of RE is that healthcare, as a complex sociotechnical system [5] is defined not only by the boundaries of physical locations.[6] People with their skills and relationships, rules,

regulations and even work place culture may also impact system performance.[5] Research into resilience of healthcare systems suggests that safety emerges from the fluid arrangement of system components or 'coordination'.[7] Coordination is hereby defined as '...the deliberate and orderly alignment or adjustment of partners' actions to achieve jointly determined goals'.[8] People who perform coordinative work are central to the process through which resources and team activities are synchronised to ensure high work performance and effectively achieve 'mission goals' in a timely manner.[9] Another important point made within the RE perspective is that safety can be enhanced by a combination of structure and control on the one hand and adaptations on the other.[10] Managers are part of a balancing structure and mediate fluidity as an integral part of everyday work without even reflecting on how they do this. There is a need to understand more about how their work is done if we are to improve safety management in acute healthcare settings.[11 12]

## Strengths and limitations of this study

► The explorative ethnographic design allowed for a deeper understanding of the underspecified (hidden) work of first-line managers.
► Several iterations of data collection and analysis allowed for an initially wide and later more focused data collection that opened opportunities for the researchers to follow-up on specific findings.
► The inductive–deductive analysis allowed the researchers to follow and describe patterns of recurring codes within the whole data set.
► The first author (KH) who conducted the fieldwork is an experienced neonatal intensive care nurse which may have affected the interpretations drawn in this study.
► The explorative study design was suitable for describing a complex process without measurable clinical outcomes.

This study aims to explore how system resilience is enhanced by naturally occurring coordination performed in situ by a management team (ie, clinical coordinators, head nurses, senior medical doctors) under variable circumstances.

## METHODS
### Design and setting
This study uses an explorative ethnographic design using participatory observations and an abductive approach to capture and analyse naturally occurring coordination in situ.[13] The reason for focusing on action–interaction was to capture a deep understanding of the varying conditions under which decisions and coordination took place.[14] The Standards for Reporting Qualitative Research checklist was used to improve transparency of the research.[15]

The study took place in a tertiary level NICU with an approximate capacity of 70 cots divided over three wards located in separate hospitals. The patient mix of the three wards is dependent on local factors such as the size and risk profile of adjacent delivery wards and the availability of paediatric surgical capacity. Staffing for the high acuity patients is normally one nursing team per two patients (one nurse and one assistant nurse). Paediatricians and neonatologists are allocated over the three wards depending on availability and competence.

Each of the three wards is managed by a clinical coordinator, a head nurse and an operations manager during daytime (figure 1). The clinical coordinator performs tasks such as rostering, planning for patient flows (admissions, discharges and transports) and clinical work when

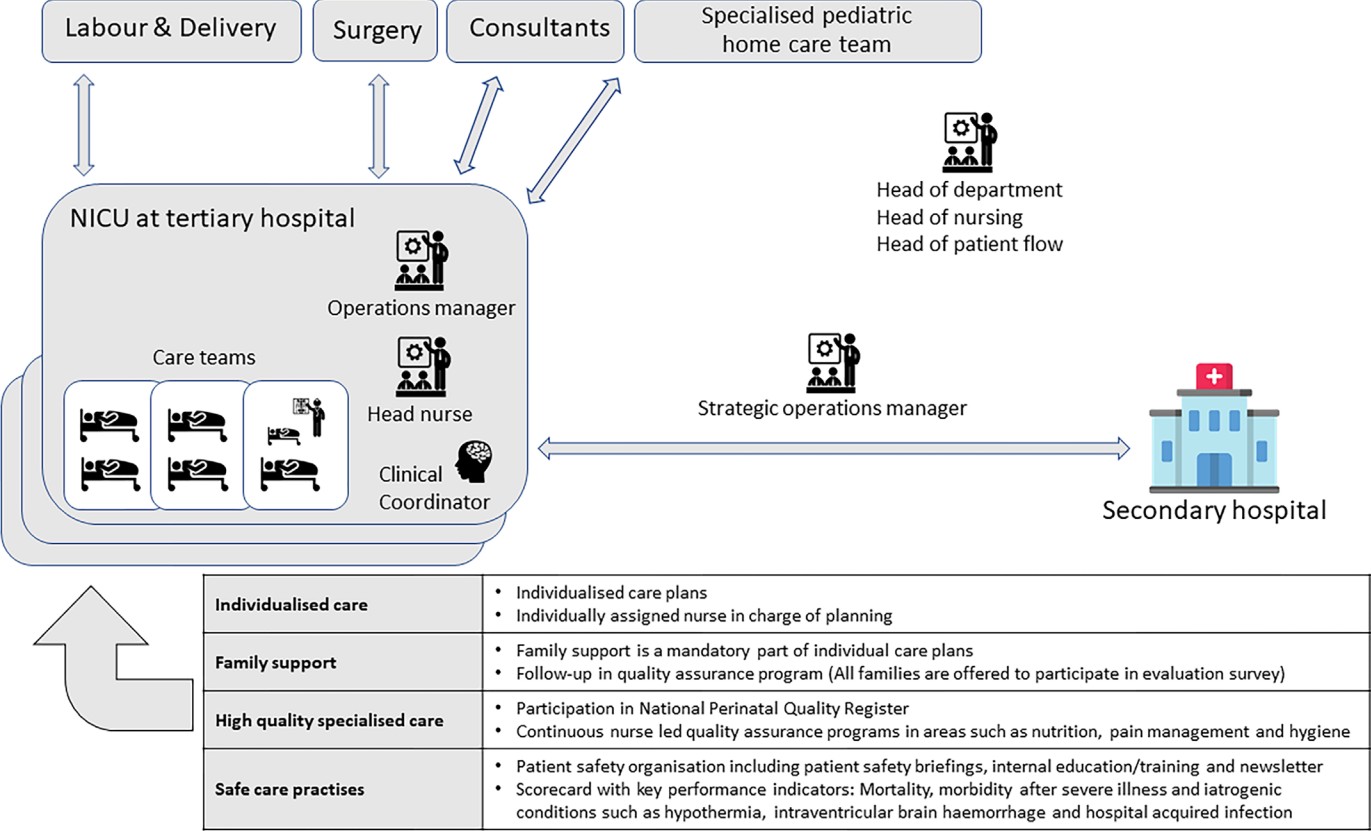

**Figure 1** The studied NICU in a tertiary level hospital with surgical capacity. Managers are visually located to illustrate their vicinity to the clinical care teams. Overarching goals are presented at the base of the figure with examples of how they are expressed in daily work. A selection of peripheral wards and units are illustrated in the surrounding area. The arrows are double headed to symbolise a two-way relationship of demands and possibilities for negotiation of for example patient transfers.

needed. The head nurse is formally responsible for the work environment and quality of care. The operations manager is a senior neonatologist with an overall responsibility for the medical quality and patient flow. A strategic operations manager has mandate to move patients between hospitals within the own organisation or to hospitals outside the region. All managers are clinical specialists (nurses and neonatologists). Hereafter, we will refer to this team as the management team.

## Data collection

Data were collected between January and February 2017. The head of the department gave permission to conduct participant observations at the unit. All staff was informed about the procedures in a staff meeting and they were informed that they could decline from being observed at any time according to the ethics approval. The clinical coordinators (four individuals) gave written consent to being shadowed after receiving written and oral information. The coordinators were all women, experienced nurses with more than 10 years of NICU experience.

Data collection was structured in iterative cycles of collection and analysis, starting with descriptive observations to get familiar with the work environment and relevant aspects of the managers' work.[16] The descriptive observations focused primarily on 'places' were the coordination of work was apparent, for example at the head nurses and secretaries open office area and at the management teams' office at the centre of the ward. The intermittent recording and analysis of field notes during initial observations yielded research questions that were in focus during the following observations.

The focused observations targeted selected situations such as rostering for the next shift, start-up meetings and handovers between shifts in addition to shadowing of coordinators. Data were collected through ad-hoc interviews with staff and managers during or after the shadowing.[16] Shadowing meaning 'following people, wherever they are, whatever they are doing'.[17] Artefacts, including coordinators' notes on patients' medical status, occupancy and rostering charts were copied and collected. All relevant aspects of the environment were captured in field notes during or after the observations, along with researcher's memo-writing over personal reflections and thoughts about what was happening.

## Analysis

All meetings and interviews were audio recorded and transcribed verbatim. Transcriptions from tape-recorded dialogues were placed in a correct temporal order along with the field notes, so that the mix of field notes and transcription chronologically represented the full workday. Transcripts were read through several times, followed by discussions in the research group about the level of detail in the data and reflections KH had regarding the observed work shift.[18] The initial inductive analysis went through a two-step process (columns two and three in table 1). First, a conversation or a situation of relevance for the study's aim (a meaning unit) was selected and the question 'What is happening here?' was directed to the data. Next, the selected conversation or situation was analysed in the context of the entire scene where it took place, with the question 'Why or how is it happening here?'. The interpretation was condensed and labelled with one or several codes (table 1). The codes and their relations were frequently discussed in the research team and sorted into tentative subcategories and main categories. Field notes were included in the analysis as means to reflect on the researchers' preunderstanding of the context. Moving back and forth between induction and deduction was a way to discover meaningful underlying patterns that made it possible to integrate concrete behaviour and

**Table 1** Example of the analysis in two-steps: going from raw data to interpretation of incident and to analytical interpretation in context, code and memo

| Raw data | Interpretation of incident | Analytical interpretation in context | Code | Memo |
|---|---|---|---|---|
| The observation begins in the flow room. The flow room is a small room with two workstations where the coordinator has her seat. There are several information sources hanging on the walls and post-it notes on the computers. The coordinator meets the observer in the flow room after having walked around the ward to check all the patient rooms. | In the physical environment, several different communication tools are gathered within a small area, that is, tools used to summarise, remember and disseminate important information. Information exchange occurs on paper notes stuck to computers, through software and when the coordinator herself walks around the ward. | Information exchange is one part of coordination and can be performed through predefined channels and tools, but also more intuitively through physical meetings within the ward while the manager compares notes to what is experienced. The environment has been adapted for having several different information channels intersecting in one place. | Adapting environment—cluster tools for facilitating information exchange. | The coordinator has gathered information about the current situation in the patient rooms by walking around and taking a look. She then looks at the system level, occupancy lists and information from the other coordinators. If coordination means exercising control, real-life information gathering is probably an important step. |
| The coordinator sits down at a computer and begins counting patients on her paper copy of the occupancy list and in TakeCare (electronic health record). She reports the occupancy in Belport (national occupancy chart) and talks a bit about this. | Counting patients and manually entering the number into the national occupancy report is one of the first things the coordinator does in the morning. | The coordinator gets an idea of the status and distributes it to the rest of the country as information. This is proactive management, as future coordination may become easier if reference can be made to Belport, or if you know that the other units in the country your information. | Information handling. | Reporting your status is a way of exercising control by impacting others perception regarding the situation at your end. |

**Table 2** The main categories on the y-axis are *actions to meet the actual situation* and on the x-axis; *operational stress*

| Operational Stress | | | |
|---|---|---|---|
| Actions to meet the actual situation. | Reorganisation | Stretching the system to work outside ordinary conditions | Routine activities under ordinary conditions |
| | Situations with several problematic trade-offs and reactive managerial work to solve immediate needs and protect high-level goals. | Adaptation of activities for optimising basic care practices using trade-offs and low-level goal sacrifices in situations of low predictability. | Situations characterised by low stress and high control with deliberate adaptations of administrative routines and work tasks to predict future situations. |
| | 'So (the other hospital) have 20 babies now, seven acute. They have opened a temporary room but have no staff for it. We have no transport team (to use as staff) because they are on their way to (another city) (…) I still won't send babies to other counties, which we can't because we have no transport team available (…) either we find some staff here that can go there or we have to order two on overtime. How do we usually do?' (Operations manager). | 'We have put those two (patients) together in 9:1 to get an emergency cot in 9:2 for the twins (9 refers to a room divided in two sections equipped for one patient per section 9:1 and 9:2), I told the father that they may have to move out. But then we will be in the situation where they maybe… They may need… That father is very new. So, I think that in that case they will have to be two (staff) in there too' (Clinical coordinator) (Q1—online supplemental appendix 1). | 'Oh, so we are only at eight intensive care babies and three in family rooms. So, it's a pretty good situation. And we have nothing acute in the delivery- or antenatal wards. I just checked' (Operations manager). |
| Supporting system cohesion | ▶ Delay work and evaluate the situation.<br>▶ Isolate problems and focus on recreating manoeuvrability.<br>▶ Exploit possibilities of extraordinary individual achievements (trade-off individual resilience for system control). | ▶ Goalsetting towards protecting manoeuvrability.<br>▶ Goalsetting for promoting basic safe care practices at the clinical level through minimal staff allocation and skill mix.<br>▶ Sacrificing continuity in patient assignments for saving lives. | ▶ Goalsetting towards family-centred care.<br>▶ Goalsetting of individualised care.<br>▶ Managing optimal staff allocation for maintaining professional development and education.<br>▶ Controlling occupancy and redundant capacity through predefined strategies.<br>▶ Monitoring state of the ward at the clinical level by regular walkarounds in the clinical work environment. |
| Extending system boundaries. | ▶ Identifying novel use of any existing external resources (ie, the use of paediatric emergency transport team and other wards).<br>▶ Shedding managerial tasks for participating in clinical emergency work (trade-off management for clinical work). | ▶ Managing occupancy trade-offs between facilities and staffing (higher occupancy in fewer rooms lowers staff requirements).<br>▶ Utilising individual managers social networks within predefined limits for proactive problem-solving. | ▶ System working within normal boundaries. |
| Adapting the structure and roles of the coordinating management team. | ▶ Make *loss of control* explicit in the management team.<br>▶ Moving from understanding the situation to making rapid decisions close to the clinical level. | ▶ Relaying information on the patients' clinical situation to mid-level and upper-level managers (information priority bottom-up). | ▶ Relaying high-level plans to clinical level workers through regular briefings (information priority top-down).<br>▶ Participate in regular medical discussions on patients' status. |
| Shifting between information sources for better sensemaking. | ▶ Dropping computerised aides, rostering systems and staffing charts for handwritten notes and memos.<br>▶ Using face-to-face communication with people in close vicinity (shedding electronic communication). | ▶ Seeking ad-hoc meetings within the management team for calibrating information of the situation and possible workarounds.<br>▶ Verbally explaining situations to other managers. | ▶ Regular use of computerised systems and handwritten notes. |

Four subcategories on the y-axis describe strategies (ie, the work that managers do); and the three subcategories on the x-axis describe the progression of a perceived level of operational stress based on the expressed availability of degrees of freedom with illustrative quotes.

deep contextual structures. Finally, a deductive comparison of interdependencies between the main categories and subcategories in relation to the theoretical concept of resilience was performed.[14]

### Patient and public involvement
Public and patient involvement was not applicable in this research.

### RESULTS
The analysis resulted in six subcategories and two unifying main categories (table 2).

About one hundred hours of focused participant observations of conversations, tasks, meetings and artefacts were performed during January and February 2017. Four clinical coordinators were shadowed throughout their shifts. Data were gathered through ad-hoc interviews with additional ten staff during the observations; fieldnotes were also collected. The analysis resulted in seven subcategories and two unifying main categories that illustrate the functional relationship between managers' *actions to meet the actual situation* and *operational stress.*

'Operational stress' describes a progression of situations that require increasingly powerful adaptive responses from a managerial team. This main category is organised from an observed baseline of activity in the sub-category *Routine activities under ordinary conditions,* which can shift to *Stretching the system to work outside ordinary conditions* and, eventually, *Reorganisation.* Activities under 'Reorganisation' were temporary, short, and resulted in bouncing back to actions in the 'Stretching' category.

'Actions to meet the actual situation' encompasses the sub-categories *Supporting system cohesion* and *Extending system boundaries,* which describe the activities performed jointly by managers and clinical staff. *Adapting the structure and roles of the coordinating management team* and *Shifting between information sources for better sensemaking* describe the management team's internal work (table 2).

Extracts from the field notes and conversations are presented to clarify the findings. More comprehensive material is provided in online supplemental appendix 1.

## Operational stress

Managers face situations where they must balance a limited number of staff with demands for high occupancy and the task of supporting clinical teams in their care of patients. Unpredictable factors such as acute admissions, staff availability and the medical progression of patients put managers in situations where they might quickly have to readjust and replan, often ad-hoc with scarce information of the overall situation. Part of the observed dialogue relate to the management team's efforts to identify alternative ways forward.

The head nurse acknowledges that the inflow of emergency patients is a top priority that is sometimes impossible to avoid.

'The patient flow, do you participate in that?' (Observer). 'No, no more than that I can say "stop", because I don't have the staffing' (Head nurse). 'Ok, so you can say that too?' (Observer). 'Mm, I can say that I have five or six teams. But…' (Head nurse). 'But you cannot say stop today' (Coordinator). 'No, but it is like that. Even if we say stop the babies are being born. And we have to take care of them' (Head nurse).

## Routine activities under ordinary conditions

The management team have planned meetings; they sweep the ward to collect information on the state of things and relay high level plans to workers.

The coordinator begins her shift by conducting a walkaround of the ward, saying hello to the nurses and doctors as she is asking if everything is all right. After the walkaround she sits down to begin exploring her staff roster and patient occupancy charts. (Field-notes)

Under ordinary conditions, there is a minimal observable need for managers to manually adapt information they extract from technical systems, regarding for example occupancy, patient acuity levels and staffing. Managers describe how their experience of the technical systems' limitations are learnt on the job and how workarounds are taught between individuals.

The coordinator makes a note of something on her occupancy chart (Field-notes). 'What did you write on that chart?' (Observer). 'CT 11:15 (Computer Tomography at 11:15)' (Coordinator). 'And it is used for that kind of information?' (Observer). 'I print one of these charts because there is more space to write on. I think… I think many… I think NN and NN (two other coordinators) use this too. And NN (another coordinator) use it because I taught her. And I use it because NN (a coordinator who quit earlier) taught me' (Coordinator).

## Stretching the system to work outside ordinary conditions

The management team must reorganise and adapt to stretched conditions for one patient while the other patients are unaffected, being cared for by their respective teams and therefore avoiding exposure of the emerging crisis that is handled by the paediatric emergency transport team.

The plan for this day is to admit one intermediary level patient that was born during the night and is waiting for a cot at the NICU. One patient has been diagnosed with a multi-resistant bacterium and will need increased hygiene standards (Field-notes) (…) 'We are plenty of people today, which is nice. But when NN (strategic operations manager) asked if we had a lot of capacity, I had to say no' (Clinical coordinator). (…) '…one important thing. This baby that needed eye surgery is now acute and will arrive soon. They will land directly in the operating room and the transport team will take care of the baby until it can go back home to its own hospital' (Operations manager) (Q2—online supplemental appendix 1).

The plan for postoperative care after acute eye-surgery was for the patient to be assigned a cot and a nursing team on the ward. The situation was managed by using the transport team to temporarily care for the baby inside the operating room until it was stable enough for transport to another hospital. This decision had possible implications for the whole management team. The clinical coordinator wanted to know about the utilisation of staff and facilities. The head nurse wanted to know about workplace safety, quality of care and economy. The operations manager had responsibility for the medical quality and the strategic operations manager for the possibility of helping one of the other wards with staffing.

## Reorganisation

The category of 'reorganisation' is characterised by managers shedding managerial tasks for participating in clinical emergency work, isolating problems and focusing on recreating manoeuvrability. The focus is to protect clinical teams from being exposed to the rapidly shifting plans and priorities at the managerial level. Strategies for regaining control can be to sacrifice the goal of continuity by transferring at-risk mothers to other hospitals (deferral), or to temporarily transfer additional weight of medical care to neighbouring subsystems such as the paediatric emergency transport team or the managers themselves.

'This is not good, its full (the ward). We have no space when this eye baby arrives' (Clinical coordinator). The clinical coordinator walks to the room where a nurse oversees the twins that were supposed to be transported out the next day but are now showing symptoms of infection (Field-notes). 'Do you think these twins could be together in a twin cot?' (Clinical coordinator). (…) 'Well, I don't know. This one is just on the margin of managing without incubator' (Nurse). (…) 'The problem is that I don't have staff

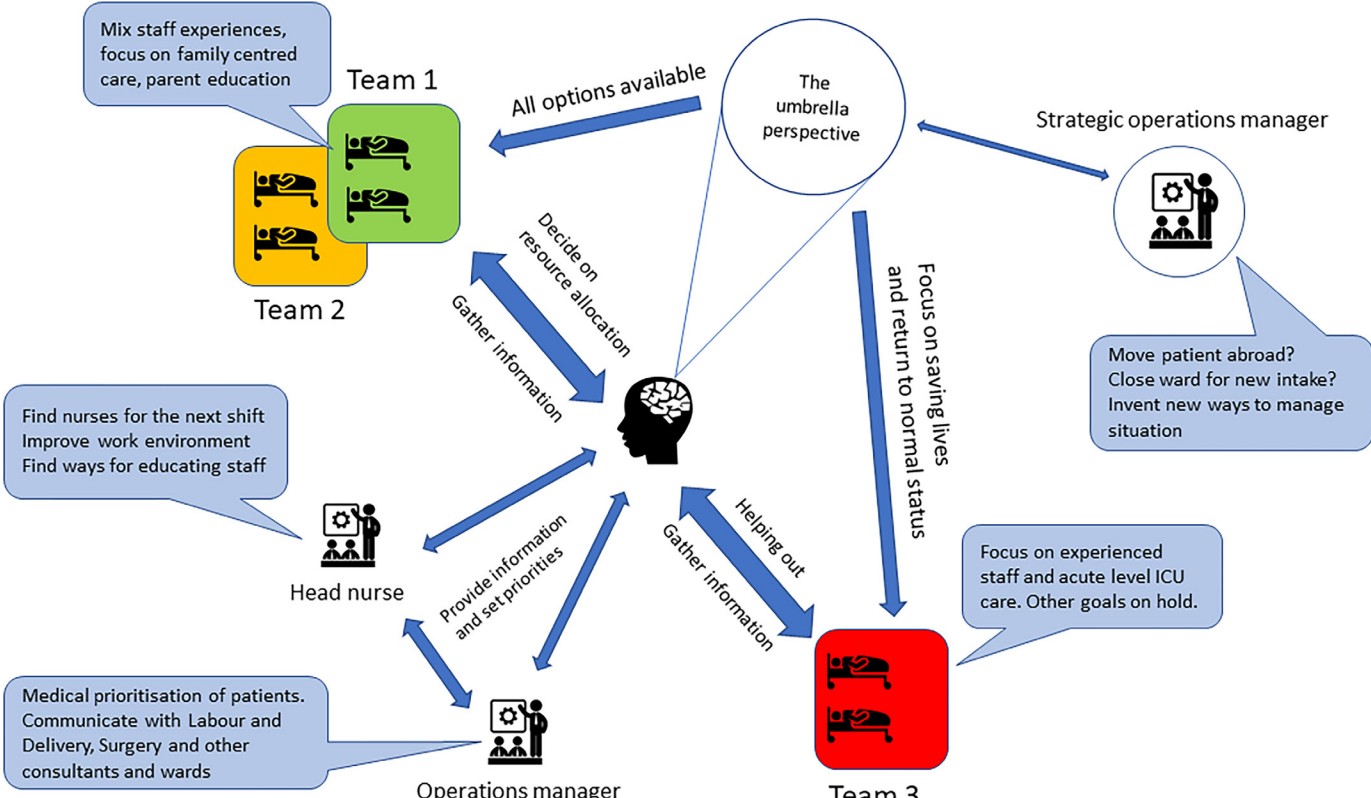

**Figure 2** Illustration of the work of balancing goal-settings for system wide coherence, maintaining the umbrella perspective and streamlining communication to meet rapidly changing demands with the shadowed clinical coordinator at the centre. The width of the double headed arrows visualises an estimation of the most frequently observed communication for the respective manager.

to open another room' (Clinical coordinator) (Q3—online supplemental appendix 1).

The manager handled the situation by putting twins together in one cot, thereby utilising one nurse to care for three babies which is more than the goal of two babies per nurse. This manoeuvre created an opportunity to temporarily handle five patients (with one empty emergency cot) in a room with staffing for four. The nurse expressed concern for her patient but remained focused on finding a solution.

When the managers started shedding managerial tasks for bedside operative work, they risked losing the ability to meet other managers and keep up to date with the ward. Management decisions in these situations were then based on a narrower understanding of the bigger picture. Computer aides were less used (or not at all) because of their inability to present rapidly changing borderline conditions.

### Adjustments to meet the actual situation

The mandate for making decisions was distributed across the management team and clinical nursing teams (speech bubbles in figure 2). The clinical coordinator moved freely across the unit and made independent decisions regarding resource allocation, rostering and whether to shed her own managerial work to help the clinical teams. The clinical coordinator also provided the other managers with information when they needed to adjust their plans.

### Supporting system cohesion

Members of the management team often communicated through ad-hoc meetings where verbal information was compared with patient rosters, patient conditions, staffing and workload indicators. They worked close to the clinical context and discussed the current goals' attainment for individual clinical teams and what the current trade-offs were. The managers compared computer-based and paper-based notes, as well as information provided at start-up meetings to update themselves of the status of the ward and where the situation was heading (umbrella perspective in figure 2).

The coordinator is back in room nine to alleviate the staff for lunch. She checks in on the baby that has been in acute surgery for intestinal obstruction. There are lots of beeping sounds, but no one seems alarmed. The head nurse enters the room and seeks the coordinators attention but initially fails as the coordinator is tending to a patient. Eventually the coordinator looks up (Field-notes). 'Didn't they (the surgeons) say already this morning that this patient was up for re-operation?' (Coordinator). 'After lunch apparently. Then what do we do with NN (another patient)?' (Head nurse).

In this example, the managers discussed how two patients' trajectories were affected by the surgeons rescheduling. The management team negotiated situations that needed simultaneous attention, like prioritising readiness and clinical capacity in some parts of the ward while maintaining family-centred care and staff education in others (Illustrated as the three clinical teams in figure 2).

### Extending system boundaries

Actions for extending the system's boundaries emerge as the pressure of prioritising decisions increase. The management team made use of other units' facilities or staff, like delaying patients in the operating theatre or letting a transport team care for the patient for some time before handing it over to the regular staff. The management team utilised auxiliary staff and overlapping competencies of different professional groups. Sometimes the managers themselves doubled as clinical staff within their vocation.

> 'If there is an acute admission and he… I do not think NN (a nurse) can have four patients by himself out there. Since the father needs quite a lot of help' (Coordinator). 'They are really good patients those babies' (Operations manager). 'What about the midwives then? (that belongs to the adjacent delivery ward)' (Coordinator). 'There are four of them? (turns to face the operations manager)' (Head nurse). 'Yesterday someone said that you can have four patients by yourself' (Operations manager). 'Not by yourself' (Coordinator). 'With an assistant nurse' (Head nurse). 'Yes with an assistant yes, that is okay but' (Coordinator). 'So there are three patients left?' (Head nurse). 'And then we have put two in the same cot at 9:1 (bed one in room nine)' (Head nurse 2). 'Okay' (Head nurse).

### Adapting the structure and roles of the coordinating management team

The management team fluently adapted its own structure in situations where there were not enough resources to manage within everyday routines, when there was urgency or when some of the management team members were not available with their specific expertise and mandate. This structural adaptation was observed when individuals in the management team shifted from relaying plans from top-down to working with patients and gathering information from bottom-up.

> In room nine is that week 22 baby that came in yesterday. They are intubating now so they use a lot of people. There were no head nurses here at seven, so I decided myself that NN (nurse) got to be alone at the stabilisation room (at the delivery ward). The doctors there have to work a little harder now. (Clinical coordinator)

### Shifting between information sources for better sensemaking

As the situation on the ward became more complex, the management team increased their reliance on handwritten notes rather than the standard computer-generated lists for staffing and patient acuity information.

The coordinator used handwritten notes as memory aides in team discussions. The notes were mainly short markings, phrases, or single words in the margins like 'discharge planned' or 'need antibiotics'. The limitations of computer-generated patient rosters to convey this type of information on real world complexities were expressed by members of the management team.

The national occupancy chart for example was only able to classify patients as high or low acuity without regarding other factors. When there was a need to work outside the binary world of two patient groups, the team stopped using this computer-generated aide and instead relied on their own domain knowledge, personal network and the stability of the management teams' understanding of the bigger picture.

The following quotes illustrates what happens when the computerised information indicated normal occupancy and the off-going night nurse reported understaffing for the same shift.

> Well and this is actually look correct (number of beds in the national occupancy chart) 14 in total. But with the stabilisation beds. That makes it 12 here plus two there… But it is also a little (inaccurate). Because then you calculate (all of them as intensive care beds). The stabilisation beds are supposed to be low acuity (Clinical coordinator).

> How did this happen? (Clinical coordinator) Well, because NN (one of the nurses) who is work-training after sick leave was included in the staffing. The parenthesis was probably put there later (points at the handwritten parenthesis in the rostering folder, indicating that NN should not be included in the staffing). (Night nurse)

## DISCUSSION

The management teams in this study exhibited a range of mindful adaptations for sustaining the units' capacity of expressing resilience. Examples include sacrificing low level goals based on up-to-date information and making continuous assessments of what would be minimally intrusive for the overall performance of the system (figure 2). As the study progressed, we came to realise that managers at the clinical level, while being central to the system's capacity for expressing resilience, did not have an explicit model or training for how they approached their work. Furthermore, managers lacked the aid of tailored decision support systems.

### Supporting coherence

The management team aimed to balance the demands and capacity of multiple care teams operating in separate

rooms while tending to patients with a wide variety of problems and levels of acuity. For practical reasons, the care teams could not always meet to communicate with each other. Small teams research performed in many modern high-risk systems - from emergency response to military, business and product development – focus on what is called a 'multiteam structure'.[19] A defining characteristic of a multiteam system is that component teams can modify their individual goal hierarchies while sharing a common distal goal or set of goals.[12] During the observations, the management team and the clinical staff agreed on making provision of acute care to rapidly deteriorating patients a top priority, allowing us to identify it as a core mission (ie, purpose of the system).[20] Other priorities were more likely to be put on hold and resumed later or to be permanently dropped. Discussions of alternate priorities in the management team could be observed in stressful situations, but the focus on providing care to the most acute patients was never questioned.

As the interdependent care teams were sometimes unable to communicate, maintaining coherence became an important factor for the managers; we describe this as the maintenance of an umbrella perspective i.e. what was needed in order to understand how the bigger picture of their interventions fit together.[21] At this point, two things can be said about coherence, with implications for the training of managers as adaptive teams.[22] First, that coordination supported a coherent goal setting with increased team collaboration and second, that it was enhanced by team members' ability to predict each others most likely priorities.

### Reorganising to support manoeuvrability under operational stress

The everyday work of the management team was characterised by seamlessly and actively organising and reorganising. Our observations revealed how the management team made use of early investments in for example staff expertise, deep domain knowledge and the workplace culture to maintain a unit wide focus on the core mission.[23] Allowing the care teams to adapt their goals individually exemplifies that being resilient is to be part of a process of identifying conflicting goals in a complex, intractable environment using 'numerous indicators in a proactive fashion to probe a system's adaptive capacity before system-wide collapse results in disaster'.[22] A realisation from studying the management team was the shapelessness of the organisation. We could not observe a formal agenda for how and why the management team was supposed to prioritise goals at the levels below the core mission. The prioritisations were consistently made in dialogue with different stakeholders such as individual nursing teams or different constellations of managers, see for example Q3 in Appendix 1. This would suggest a decision-making process that defers to social rather than hierarchical aspects of the system.

Our study suggests that it is up to the management team to support the system by using experience, professional ethos and domain knowledge to negotiate the way forward in a manner that resembles the deference of expertise, as described by Weick and Sutcliffe.[24] Specifically, because the flexible decision structures enabled resilient performance when expertise and experience outranked formal hierarchical positions. We did not receive a coherent answer to the question of how the coordinators developed their ideas on how to prioritise, since they all explained that they learned on the job (without formal training) and that they could only rarely watch each other's work.

### Balancing between long-term and short-term goals

Resilience depends on the use of earlier investments in 'potential opportunities for action' previously described as degrees of freedom.[25] However, it could be argued that sacrificing low-level goals does represent a loss of potential future degrees of freedom if this is done often. In the context of the NICU, families are less prepared for discharge if they are not trained, staff might receive less time learning from experienced colleagues if they do not work together and formal routines might erode if they are not employed.

The link between adaptation and outcome (whether successful or unsuccessful) as described for example in the CARE model for researching resilience in healthcare,[26] was important for the application of resilience in everyday clinical work. An adaptation is a deviation from work as planned, and it is not always clear beforehand whether the outcome of an adaptation constitutes success or failure in terms of quality and safety. Our study describes managers' adaptive responses to the conflicting demands of acute patient care on one hand and long-term strategic demands on the other (measured as, eg, respirator days, patient throughput and hospital acquired bloodstream infections).

Because structure and policy, as well as constant adaptations, are required to sustain operations even without disruptive events, it is not possible to deterministically describe outcomes of coordination as successful or unsuccessful. However, we have described how coordinative work contributes to the system's capacity for expressing resilience. The balancing act between seemingly irreconcilable goals makes it impossible to determine, even in retrospect, whether coordination was good or bad for the total outcome of the system. Each decision to suspend or sacrifice a low-level goal has implications for the organisation's future capacity for expressing resilience. All teams worked towards the core mission of providing acute care. However, the maintenance of long-term investments was achieved by mindfully trading off a range of low-level goals sacrificed by the clinical teams (ie, one sacrificing patient education and another sacrificing staff education).

### LIMITATIONS

This was a single-centre study of a specialised unit with a specific patient clientele that cannot be cared for by any other type of unit available. It was expected that the unit's

high tempo and specialisation would promote a particularly observable coordinative work with the risk that it might introduce the argument of limited transferability to other areas of healthcare. We believe however that the networked structure of three wards is not unique. The specific unit for study enabled us to capture and understand the subtleties of everyday work of first-line managers. Further studies are needed to investigate how much of this work may be specific to organisations of both similar and contrasting types.

Using a qualitative cross-sectional design, this study does not allow us to define successful or unsuccessful outcomes. Resilience is described based on the actions taken and further studies are required to operationalise and test our results.

The first author (KH) who conducted the fieldwork is an experienced neonatal intensive care nurse with experience from the studied NICU. The familiarity with the specific type of work may have affected the interpretations drawn in this study.

Trustworthiness during data analysis was addressed by regular peer-check and in seminars with the wider research group and member check. The final analysis was individually validated with the coordinators.[27] The iterative process of data collection and analysis was intended to ensure that the analysis included more than one researcher's interpretation. Transferability was addressed by leaving an audit trail of extracts from the data in the report so that readers from other fields can evaluate if the results are transferable to their respective contexts.[28] Table 1 of the methodological section provides a trail of how interpretations of data were made.

The use of voice recordings of meetings and conversations was limited to situations where verbal consent could be obtained from all participants, unless explicitly asked not to by any of the participants.[29] In larger groups, where participants attended only partially, this was not a feasible option and handwritten notes were taken, video recordings were not possible because of difficulties with assuring patient anonymity in the clinical context.

## CONCLUSIONS

Our findings describe a functional relationship between operational stress and actions taken to meet the actual situation. We suggest that patterns of managerial activities along a continuum of operational stress are indicative of when and how resilience emerges. Actions for enacting resilient capacity include for example quickly organised ad-hoc meetings, shedding managerial tasks, novel use of resources, and isolating and protecting selected areas of the system (depending on the circumstances). To make the findings of this study actionable, the concepts of 'operational stress'and the bundles of 'actions taken to meet the actual situation' need to befurther researched and operationalised.

We also suggest that in design of future research into coordination, the focus of assessment and reflection should be on adaptive managerial responses in situations where the system is 'stretched' or 'in need of reorganisation'. (table 2)

Furthermore, it is important that healthcare policy and organisational redesign initiated at higher levels are well calibrated with the nature of managerial work at the clinical level before any interventions are developed.

**Acknowledgements** Thank to Professor Eric Hollnagel who contributed with theoretical expertise on the interpretation of data; Professor Richard Cook for valuable discussions about system safety and methodology in the early phase of the study. Thanks also to the participating Neonatal unit.

**Contributors** The study was initiated by KH. The study design was developed in collaboration within the research team. KH performed the data collection. The analysis and interpretation of data was conducted in close collaboration between KH, ME and KPH. KH wrote the first draft of the paper. All authors contributed with writing, critical revisions and approval of the final version.

**Funding** The authors have not declared a specific grant for this research from any funding agency in the public, commercial or not-for-profit sectors.

**Competing interests** None declared.

**Patient consent for publication** Not required.

**Ethics approval** This study was approved by the regional ethical review board of Stockholm (2016/1832-32).

**Provenance and peer review** Not commissioned; externally peer reviewed.

**Data availability statement** Data are available upon reasonable request. All data relevant to the study are included in the article or uploaded as supplementary information. Given that the data is in Swedish and phrasing of the consent obtained from participants, complete raw data are not available for sharing. Partial data sets used and/or analysed during the current study are available from the corresponding author on reasonable request. A selection of translated quotes is supplied in the results and methods sections and in appendix 1.

**ORCID iDs**
Karl Hybinette http://orcid.org/0000-0002-6989-3762
Mirjam Ekstedt http://orcid.org/0000-0002-4108-391X

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
