## [Reviewer comments · BMJ Open]

ARTICLE DETAILS

TITLE (PROVISIONAL)	A FIRST LINE MANAGEMENT TEAM'S STRATEGIES FOR SUSTAINING RESILIENCE IN A SPECIALISED INTENSIVE CARE UNIT – A QUALITATIVE OBSERVATIONAL STUDY
AUTHORS	Hybinette, Karl; Pukk Härenstam, Karin; Ekstedt, Mirjam

VERSION 1 – REVIEW

REVIEWER	Christopher S Parshuram Hospital for Sick Children, Toronto, Canada
REVIEW RETURNED	16-Jun-2020

GENERAL COMMENTS	Thank you for the opportunity to review the paper by Hybinette et al. Important work that provides a description of the 'things' that we (as managers) learn to do by observation and experience. Overview: Very important, innovative work in an area of great relevance to the operations of intensive care. The authors conducted and report an ethnographic study of a 3-ICU network in Sweden, using general observation, transcribed audio-recording and qualitative synthesis. "This study aims to explore how system resilience is sustained by naturally occurring coordination performed in situ by a management team (i.e. clinical coordinators, head nurses, senior medical doctors) under variable circumstances." Broadly, (in my opinion) the addressable issues relate to the assumptions embedded within the research question (resilience is sustained), the assumptions of the 'investigating team' (related to results and interpretations); description /definition of the 'core mission' and the roles of the management team; clarity of distinction between usual and special circumstances (is adaptation actually routine work – rather than allowing established processes to continue); technical separation of manuscript components; and the generalizability of the results to other settings. I have divided my comments into major and minor (below) [1] Structure and separation. Interwoven methods & results. [a] The Introduction is long and in my opinion meanders to a crisp summary of the study purpose. I really liked the first paragraph of the results as a summary of the NICU environment and suggest this is used as paragraph one of the introduction. >& I suggest significantly shortening the introduction to <2/3 of a page.
--

[b] Methods are fragmented, could be consolidated and revised to improve the flow. The methods appear in the results and could be rearranged to better help the reader navigate what was collected and how, and then what was done with the data. For example telling the reader the results are the end of methods “Categories and subcategories were refined, and the researchers agreed upon six subcategories and two main categories that unified them” is a result.

[c] It is often a challenge to decide the best place for description of the environment / setting in methods or background or results. Perhaps this might be placed in the introduction (within the limitations suggested above). This I also note the NICU – environment here is a highly integrated network – across three hospitals. This may have implications for generalizability and the sorts of readers most interested of this article.

>It would be helpful to clarify about the three inter-related units, with organizational overlap and a mandate (I assume to collaborate for overall efficiency, rather than self-interest).

[d] better separation [bracketing] the current manuscript leaves me with the impression that the authors have interposed their experience into the data For example “The CMT need to balance the demands and capacity of multiple teams that operate in separate rooms tending to patients with a wide variety of problems and acuity” (line 313). And “When the managers start shedding managerial tasks for doing operative work at the bedside, they will lose the ability to meet other managers and keep up to date with the ward.”

[2] Mission/Roles/Scope.

[a] Explicit description /definition of the ‘core mission’ as understood by the managing team / the authors and the individual roles of the management team. Some statements suggest that the core mission is understood (“...allowing us to identify it as a core mission”). “...characterised by seamlessly and actively organising and reorganising.” & “high occupancy while supporting clinical teams in their care of patients”.

For me figure one did not sufficiently address this gap. I wondered also about the separation of the four boxes (Is high-quality specialized care not the same as safe care and individualized care ?).

>I suggest the mission should be explicitly described either as understood by the investigative team (in introduction) or as abstracted from the notes with the participants (in results). This would provide important clarity for readers to understand the decisions made / actions taken by the managers (ie. The context of/ for their observed behaviours) and to help interpret ‘prioritization’ and adaption. Perhaps this is the first part of the results, or might be added to a revised introduction (if you, the authors, indicate that mission is as understood by you).

>I suggest the roles might be placed in a table that includes scope and perhaps examples of things that are out of scope (possible examples lobbying more senior administrators for more beds, or sending people who are not needed home).

[b] Figure 2 is good, however looks restrictive / incomplete. If this is an organization structure diagram it could be more explicitly stated, and the footer include links to the table described below.

>I suggest that the authors could add new information to strengthen the readers understanding of who was studied, of responsibility (who is responsible for administrative decisions ?

	and who has authority to make the final decision), and perhaps of the volume of information from each participant type. [3] Distinction between usual and special = the work describes what is done as a routine, this is characterized as adaption, a 'special case'. I note in Table 2 that three categories are described. Everyday, stretching, and losing control. Greater clarity on what the usual role of one of the management team is / constitutes might help provide clarity on the activities. >I suggest describing these three different categories in results (separation of 'outside ordinary' vs. everyday might be clarified from the data with examples) or if defined a priori then in methods. [4] When under 'heavy stress' (what does this mean?) ... the data left me wanting more "priorities are accepted while others are discarded as unacceptable." (what was unacceptable?). >I suggest : indication of the frequency that each of the 3-levels of activity happened, and the examples might be provided for the 'choices made' / trade offs, rather than the current use of idle / under-utilized resources. [5] Results: The two main categories adaptations for enabling safe performance and maneuverability. It was not immediately clear how these are different, the inclusion of both in the table I found reinforced my sense that these concepts overlapped. Is manoeuvrability a system quality, and adaption the process of moving within it? >I suggest increasing the specificity of descriptions of these categories in the results section, adding some statements that describe the contrasts, and reviewing ./ revising Table 2, to increase the utility of information For example how do "System working within normal boundaries" and "Regular use of computerised systems and manual cognitive aides" add ? The quotations might be tabulated to give more space to describe the concepts related to what managing persons do when resources are 'tight' (or perhaps to best match needs with available resources). I note the table does not include deferral as a management strategy. To be useful I think the table / or other presentation should be comprehensive. [6] Transparent Presentation. The authors mention 86 codes, and their distillation to two main categories and 6 sub-categories. The codes, their definitions + illustrative quotes and how they were distilled into the final 4 sub-categories might be provided in an appendix. [7] Limitations. There are multiple strengths of this work (novelty, direct observation, qualitative methodology, and relevance to ICU practice) that I think render its contributions valuable. I suggest including an expanded limitations section – that includes acknowledgment of the lack of clinical outcomes / the assumption of resilience was achieved by the actions described – if quality of decisions was abstracted then this should be included, that the data provide a cross-sectional description of what is done with the goals of X, do not tell us how or what to improve, and complement the statements made about future directions. There are several smaller points.
--	---

	[a] tense ...past tense, and clarity that what was observed in the context of this study rather than a 'universal'. [b] statement intent vs assumption : line 174 : 'The researcher's involvement with clinical experts ensured that the patient perspective were in focus.'" I think this should be substantiated from the data, rephrased as the intent, or removed. My preferred option is remove. [c] potential over-specification eg ... 'cognitive aids' for (what I assume to be lists on paper and computers). [d] description of Safety-1 and Safety-2 warrants clarification (if not removed by the authors in the next revision). [e] figure 1. Above comments + suggest including clear indication of the roles that were studied in this work. [f] figure two is good. However, it serves to highlight the need to better describe the roles, authority and responsibilities of each actor – including the clinical co-ordinator at the center of the CMT. [g] abbreviations – perhaps this is me, however I think that removing these wherever possible (also known as ...Everywhere) is highly desirable. [h] overlap with non-healthcare management – might receive some attention in discussion . I thank the editors for the invitation to review this interesting and potentially formative work, and acknowledge the significant efforts to obtain and synthesize the data by the authors and their teams. I believe that with revision that this work can achieve its substantial potential, and look forward to seeing its next iteration. Sincerely Chris Parshuram
--	--

REVIEWER	Clare Crowley Oxford University Hospitals NHS Foundation Trust
REVIEW RETURNED	29-Jul-2020

GENERAL COMMENTS	This is a timely and interesting study that provides valuable new insight into healthcare resilience. The information of the relationship between necessary adaptations and systems manoeuvrability should be shared. Clearly lots of thought went into the study design and analysis. However as it is currently written it is readily accessible to the reader. If comments were addressed that this is suitable for publication, but not in its current format. Review by a native English speaker would be worth considering. To make this accessible to a reader that is unfamiliar with healthcare resilience and even some of the current thinking about safety-2, work system components more explanation of these concepts with plain English would help the reader to follow the flow. Specific comments: abstract - methods section include the preparatory work before the focussed shadowing, study setting i.e. tertiary centre in Sweden. Limitations - use of video recoding techniques with reflection e.g. VRE Has the research worked in the study setting as a practitioner? Was the researcher experienced in neonatal intensive care or just intensive care? Parents perspective or advisor to research?
--

	Background - a modern definition of safety would help with a common understanding for when safe performance is discussed later on. line 85 - what are the boundaries of the system being studied? Be clear that this is more than a physical location) line 89 - can safety really be "managed" - maybe created or enhanced? line 97 - explain what 'partners' means in the healthcare setting Is the study aim to explore the role or contribution of co-ordination to creating healthcare resilience Methods - would the SRQR guidance help ensure all the required elements are included? What was sampling strategy, participant recruitment and consent? Was software used to assist with the content analysis? line 122 - three rather than tree wards? table 1 - what is Belpoit is this a ward name or a place? PPI - the research cannot provide a patient or public perspective. If this was not a part of the study just need to be clearly stated. Table 2 - what is PETS? Line 341 - what is (a.a)? How does the managers contribution to resilience link with other enablers of resilience?
--	--

VERSION 1 – AUTHOR RESPONSE

Reviewer: 1	
[1] Structure and separation. Interwoven methods & results. [a] The Introduction is long and in my opinion meanders to a crisp summary of the study purpose. I really liked the first paragraph of the results as a summary of the NICU environment and suggest this is used as paragraph one of the introduction. >& I suggest significantly shortening the introduction to <2/3 of a page.	Thank you for your advice. We have moved the first paragraph of the results to the introduction and shortened the background accordingly. Figure 1 has been moved to to background (page 4) for clarity.
[b] Methods are fragmented, could be consolidated and revised to improve the flow. The methods appear in the results and could be rearranged to better help the reader navigate what was collected and how, and then what was done with the data. For example telling the reader the results are the end of methods “Categories and subcategories were refined, and the researchers agreed upon six subcategories and two main categories that unified them” is a result.	The methods have been restructured under more specific headlines (Design and settings, data collection, analysis) and revised for any misplaced results, discussion and ethical considerations.

[c] It is often a challenge to decide the best place for description of the environment / setting in methods or background or results. Perhaps this might be placed in the introduction (within the limitations suggested above). This I also note the NICU – environment here is a highly integrated network – across three hospitals. This may have implications for generalizability and the sorts of readers most interested of this article. >It would be helpful to clarify about the three inter-related units, with organizational overlap and a mandate (I assume to collaborate for overall efficiency, rather than self-interest).	The part of the settings description that relate to figure 1 is moved to the introduction as suggested and expanded with a short paragraph of the network- and goal structure. Figure 1 has been updated with the sources of the overarching goals (boxes at the bottom), this revision also apply for comment 2 (a). (Page 4)
[d] better separation [bracketing] the current manuscript leaves me with the impression that the authors have interposed their experience into the data For example “The CMT need to balance the demands and capacity of multiple teams that operate in separate rooms tending to patients with a wide variety of problems and acuity” (line 313). And “When the managers start shedding managerial tasks for doing operative work at the bedside, they will lose the ability to meet other managers and keep up to date with the ward.”	Thank you for pointing this out. The results have undergone a major revision. Language have been revised to better reflect when we observed behaviour and refer to concrete exclamations. To improve transparency, the results section have been expanded with more data (quotes) and clear referencing to either (Field-notes) or the title of the person quoted. The term CMT has been completely removed in favour for the simpler term “management team”.
[2] Mission/Roles/Scope. [a] Explicit description /definition of the ‘core mission’ as understood by the managing team / the authors and the individual roles of the management team. Some statements suggest that the core mission is understood (“...allowing us to identify it as a core mission”). “...characterised by seamlessly and actively organising and reorganising.” & “high occupancy while supporting clinical teams in their care of patients”. For me figure one did not sufficiently address this gap. I wondered also about the separation of the four boxes (Is high-quality specialized care not the same as safe care and individualized care ?). >I suggest the mission should be explicitly described either as understood by the investigative team (in introduction) or as abstracted from the notes with the participants (in results). This would provide important clarity for readers to understand the decisions made / actions taken by the managers (ie. The context of/	Line 76-78 - The mission is now properly described as understood by the authors (with reference to source data in figure 1). Figure 1 is updated with examples of sources for each of the overarching goals and the figure legend is updated accordingly. (Page 4)

for their observed behaviours) and to help interpret 'prioritization' and adaption. Perhaps this is the first part of the results, or might be added to a revised introduction (if you, the authors, indicate that mission is as understood by you).	
>I suggest the roles might be placed in a table that includes scope and perhaps examples of things that are out of scope (possible examples lobbying more senior administrators for more beds, or sending people who are not needed home).	Line 258-262 - Introduced an additional paragraph to help clarify the interpretation of fig.2 the roles and relationships regarding decision making between the management team and clinical teams. We have observed a flexible decision structure that is discussed on line 363-367.
[b] Figure 2 is good, however looks restrictive / incomplete. If this is an organization structure diagram it could be more explicitly stated, and the footer include links to the table described below. >I suggest that the authors could add new information to strengthen the readers understanding of who was studied, of responsibility (who is responsible for administrative decisions ? and who has authority to make the final decision), and perhaps of the volume of information from each participant type.	Page 16 - Added a short paragraph above the figure to address the issue of responsibility (interpreted as mandate). Arrows are made with different thickness to indicate the observed volume of communication between respective people. The legend is updated. The figure moved to the top of main category 2 in results for better reading flow.
[3] Distinction between usual and special = the work describes what is done as a routine, this is characterized as adaption, a 'special case'. I note in Table 2 that three categories are described. Everyday, stretching, and losing control. Greater clarify on what the usual role of one of the management team is / constitutes might help provide clarity on the activities. >I suggest describing these three different categories in results (separation of 'outside ordinary' vs. everyday might be clarified from the data with examples) or if defined a priori then in methods.	Page 11 - Added quotations for each of the suggested sub-categories in table 2 for clarification and revised the "Manouverability" section in the results. Revised the names of the main category on the y-axis for clearer distinction. Updated the first paragraph after table 2 for better clarity.
[4] When under 'heavy stress' (what does this mean?) ... the data left me wanting more "priorities ... are accepted while others are discarded as unacceptable." (what was unacceptable?). >I suggest : indication of the frequency that each of the 3-levels of activity happened, and the examples might be provided for the 'choices made' / trade offs, rather than the current use of idle / under-utilized resources.	We thank you for this suggestion, we added a table-3 on page 12 with some tabulated quotes to illustrate problematic situations and their respective choices/trade-offs for each of the manouverability categories. Unfortunately we cannot provide a reliable estimation on the frequency of the three levels

	of activity because of the qualitative study design. The purpose of the study are to describe the continuous process of managing. Our method does so far not allow for quantification. However we plan to continue this study by operationalising some of the observed strategies and explore the possibility of continuously measuring the sustainability of the system and look for relevant early warning for when it is near an unacceptable situation/erosion.
[5] Results: The two main categories adaptations for enabling safe performance and maneuverability. It was not immediately clear how these are different, the inclusion of both in the table I found reinforced my sense that these concepts overlapped. Is manoeuvrability a system quality, and adaption the process of moving within it? >I suggest increasing the specificity of descriptions of these categories in the results section, adding some statements that describe the contrasts, and reviewing ./ revising Table 2, to increase the utility of information For example how do “System working within normal boundaries” and “Regular use of computerised systems and manual cognitive aides” add ? The quotations might be tabulated to give more space to describe the concepts related to what managing persons do when resources are ‘tight’ (or perhaps to best match needs with available resources). I note the table does not include deferral as a management strategy. To be useful I think the table / or other presentation should be comprehensive.	The results have undergone major revision with this suggestion in mind, Table 2 are revised with additional quotes for the manoeuvrability sub-categories and additional revisions of text as described in point 3 above. Additional illustrative quotes have been added to further exemplify the results. Table-3 might also add further information. There was a description of deferral on line 238, revised the wording for clarity.
[6] Transparent Presentation. The authors mention 86 codes, and their distillation to two main categories and 6 sub-categories. The codes, their definitions + illustrative quotes and how they	All data and analysis are in Swedish, due to the comprehensive work of translating we have opted to provide additional data in appendix 1 and more upon reasonable request. For this

were distilled into the final 4 sub-categories might be provided in an appendix.	revision we provide additional quotes and explanations and hope that will suffice.
[7] Limitations. There are multiple strengths of this work (novelty, direct observation, qualitative methodology, and relevance to ICU practice) that I think render its contributions valuable. I suggest including an expanded limitations section – that includes acknowledgment of the lack of clinical outcomes / the assumption of resilience was achieved by the actions described – if quality of decisions was abstracted then this should be included, that the data provide a cross-sectional description of what is done with the goals of X, do not tell us how or what to improve, and complement the statements made about future directions.	We have added a paragraph in the revised “limitations” to address this issue (Line 400-402).
There are several smaller points.	
[a] tense ...past tense, and clarity that what was observed in the context of this study rather than a ‘universal’.	The results was revised with this in mind.
[b] statement intent vs assumption : line 174 : ‘The researcher’s involvement with clinical experts ensured that the patient perspective were in focus.’ I think this should be substantiated from the data, rephrased as the intent, or removed. My preferred option is remove. (Removed)	The formulation was removed PPI also according to comments from reviewer 2.
[c] potential over-specification eg ... ‘cognitive aids’ for (what I assume to be lists on paper and computers).	Replaced with “handwritten notes”.
[d] description of Safety-1 and Safety-2 warrants clarification (if not removed by the authors in the next revision).	Terminology removed in revision for better readability.
[e] figure 1. Above comments + suggest including clear indication of the roles that were studied in this work.	Revised as described in above comments.
[f] figure two is good. However, it serves to highlight the need to better describe the roles, authority and responsibilities of each actor – including the clinical co-ordinator at the center of the CMT.	Figure 2 is revised according to above made comments. Some clarification in the results that refer to the figure. The figure changed place to line 263 in the results for a more logical flow.

[g] abbreviations – perhaps this is me, however I think that removing these wherever possible (also known as ...Everywhere) is highly desirable.	Revised the entire document, abbreviation CMT is completely removed. ECW also.
[h] overlap with non-healthcare management – might receive some attention in discussion .	Line 363-367 added a link of the resemblance between hospital managers work to general management theory (HRO) in discussion.
Reviewer: 2	
Specific comments: abstract - methods section include the preparatory work before the focussed shadowing, study setting i.e. tertiary centre in Sweden.	The methods section (Data collection) have been revised for consistency regarding the order of first “Descriptive observations” and then “focused observations” have been clarified.
Limitations - use of video recoding techniques with reflection e.g. VRE	Addressed in the Limitations section. Line 413-417
Has the research worked in the study setting as a practitioner?	Made clarification in the Limitations section Line 403-405
Was the researcher experienced in neonatal intensive care or just intensive care?	Yes, made clarification in the Limitations section and specified in the bullet point in “strength and limitations”
Parents perspective or advisor to research?	We are not sure about this comment, but the parent perspective is not relevant for this project with a focus on the hospital system.
Background - a modern definition of safety would help with a common understanding for when safe performance is discussed later on.	Definition provided in the introduction Line-81-82.
line 85 - what are the boundaries of the system being studied? Be clear that this is more that a physical location)	Clarified on boundaries in the background Line 87-89.
line 89 - can safety really be "managed" - maybe created or enhanced?	Line 86 - We revised the wording to “enhanced” to also signify our understanding that safety is always present in some degree, that we aim to make a system safe(r).
line 97 - explain what 'partners' means in the healthcare setting	Line 92-93 Added a paragraph to clarify our standpoint.
Is the study aim to explore the role or contribution of co-ordination to creating healthcare resilience	Yes, how coordination contributes to sustain resilient performance of the healthcare system. We hope that the formulation in the aim and abstract is made clearer with the rephrasing of

	“sustained” to “enhanced” and revision of the discussion.
Methods - would the SRQR guidance help ensure all the require elements are included?	Yes, thank you for pointing this out, it is used and now stated so in the methods section. Line 108-109
What was sampling strategy, participant recruitment and consent?	Clarification added under “data collection” section Line 125-130.
Was software used to assist with the content analysis?	We have not used any specific program such as INVIVO or MAXQDA.
line 122 - three rather than tree wards?	Thank you for pointing this out, fixed
table 1 - what is Belport is this a ward name or a place?	The name is revised and is now referred to as “National occupancy chart”
PPI - the research cannot provide a patient or public perspective. If this was not a part of the study just need to be clearly stated.	Adjusted according to comment.
Table 2 - what is PETS?	Paediatric emergency transport system, the abbreviation has been replaced with the full name.
Line 341 - what is (a.a)?	(a.a) is a leftover Swedish abbreviation from an earlier draft. Removed.
How does the managers contribution to resilience link with other enablers of resilience?	A link to management theory provided in discussion. Line 363-367.

VERSION 2 – REVIEW

REVIEWER	Christopher S Parshuram Hospital for Sick Children, Toronto. Canada
REVIEW RETURNED	20-Sep-2020

GENERAL COMMENTS	Thank you for the opportunity to re-review the revised paper by Hybinette et al. The authors provide rationale for improved understanding of resilience ‘structure and control on one hand (i.e. Safety -1) and adaptive behaviour’ describe their focus on co-ordination by managers. Data are from observations over a 2 month period (Jan & Feb 2017) from a 3-unit group of NICU’s in Sweden. My impression is that the study findings were to describe the context for managerial (aka co-ordination) activities – (usual, emerging, & stressed) and examples of observed activities. Guiding principles
---

for decision-making beyond the variably articulated 'core mission' did not emerge from the analysis.

I arrange my comments into major and minor below:

Major comments.

[1] Introduction:

There remains opportunity to increase the focus of this section as it leads us to the study conducted.

[a] The generalizable value of the work (from my perspective as a healthcare professional who does not work in an NICU) is about the study of managers of a complex environment. Specifically: That (effectively) achieving 'mission goals' in acute healthcare environments requires system resilience. That resilience is the combination of organizational structure and fluidity. That the balance of structure and fluidity requires co-ordination that is mediated by managers.

> Accordingly, I suggest

[a] Edit Description of the environment – I recall suggesting it might be moved into the introduction – and on re-reading the revised text I think the description provided in the methods is strong, and with the figure referenced from the methods section is ample. Thus paragraphs one and two of the introduction might be removed.

[b] The authors might then consider opening with a modification of the paragraph that starts on line 80. "In acute care environments maintaining quality and safety is a persisting challenge..." I suggest the NICU environment might then be described in a 1 or 2 sentences that confirm that NICU provide acute care, and NICU environments are characterized by complexity, production pressure and need for co-ordination.

[c] Line 97-99 talks about how managers learn ... while this may be important it seems tangential to the goal of a study "...to explore how system resilience is enhanced by naturally occurring co-ordination..." Perhaps this section could be edited out.

Methods: no new comments

Results: I think there are opportunities to improve the flow and internal logic of the results.

[2] Context: The results begin abruptly. In a revision they might begin with the time frame and some more explicit description of the participants, and the number approached (to give a sense of the representativeness of the sample studied) who was studied, and the volumes of data collected ... I found the immediate 'look somewhere else' was disruptive – as the authors directed me (the reader) to tables / other parts of the manuscript.

[3] Labels: For me the 'manoeuvrability' label does not fit well with what I think it describes: the situations that may 'require' adaption or perhaps (my preference) describe degrees of operational stress. I appreciate the conceptual model of structure and flexibility articulated as rationale for the study. From [i] usual comfortable 'ok' activity (baseline), to [ii] recognizing increasing stress on the available resources (problem recognition and standard responses) then to [iii] modifying usual practice / special adaptations/responses. I might be missing something about this label that others can see. Greater clarity about these 'big categories' –ideally presented early in the results- may help other readers and provide links back to the rationale for flexibility mentioned in the introduction.

[4] Adjustments in Usual. Table 2: As I reflected on this important table I wondered about the rationale that 'adjustments' are needed

	for 'everyday work'. Perhaps this is a semantic / definitional issue, however to me it would seem that any adjustments would be from this 'usual' baseline. Reviewing the table I think the content of the cells in this column (the usual baseline situation) is either indirectly linked to resilience or might be described as the doing of mission-relevant activities. The authors may consider that a more inclusive description is that these columns represent activities rather than adjustments, and consider re-ordering the columns to reflect the transition from usual activities to special responses. [5] Core Mission: Interpretations about the (perhaps shared or divergent) understanding of 'core mission' (to inform operational priorities) by participants would strengthen the interpretation of the activities taken across the degrees of 'operational stress' from usual to special actions. [a] I note in the discussion... 'agreed on making provision of acute care to rapidly deteriorating patients a top priority, allowing us to identify it as a core mission'. The statements about 'umbrella perspective' refer to "team members' ability to predict the most likely priorities of each other." Greater description of data/ observations about this 'ability' would strengthen the work. [b] As I write this I wonder did the participants agree on these priorities (was this a shared mental model or an understanding of different mental models?). Were there observed moments of disagreement reflecting tensions between the interests/ priorities of the multiple teams mentioned. [c] the statement : "we could not observe a formal agenda for how and why the management team was supposed to prioritize in terms of goal achievement below the core mission" may have implications for the 'formal' training of future managers, and suggests to me that individualized management /practice / co-ordination responses were common. Were participants asked about this more detailed prioritization? What did they say? [d] I think the measurement of successful decision-making / co-ordination comes back to the mission.... Was it achieved. The authors might reflect more on this if it aligns with the emergence of new data. [6] Conclusion: the first sentence seemed redundant, and other statements 'central to the capacity for expressing resilience' are (in my opinion) not well connected to the data. The final sentences about interventions and defining successful co-ordination are important, but diverge from the study findings. I suggest revising to reflect findings and gaps to be addressed in future work. [7] Discussion: might be expanded to better integrate understanding from other industries Smaller points [a] Table 3 might be added to table 2. [b] The figure... perhaps decreasing the color intensity of the blue speech bubbles – I found they distracted from the main information at the center of the figure. I enjoyed reading this revised manuscript and hope that my comments and suggestions aid the authors and editors in their quest to improve
--	---

VERSION 2 – AUTHOR RESPONSE

Major comments.

[1] Introduction:

There remains opportunity to increase the focus of this section as it leads us to the study conducted.

[a] The generalizable value of the work (from my perspective as a healthcare professional who does not work in an NICU) is about the study of managers of a complex environment. Specifically: That (effectively) achieving ‘mission goals’ in acute healthcare environments requires system resilience. That resilience is the combination of organizational structure and fluidity. That the balance of structure and fluidity requires co-ordination that is mediated by managers.

> Accordingly, I suggest

[a] Edit Description of the environment – I recall suggesting it might be moved into the introduction – and on re-reading the revised text I think the description provided in the methods is strong, and with the figure referenced from the methods section is ample. Thus paragraphs one and two of the introduction might be removed.

[b] The authors might then consider opening with a modification of the paragraph that starts on line 80. “In acute care environments maintaining quality and safety is a persisting challenge...” I suggest the NICU environment might then be described in a 1 or 2 sentences that confirm that NICU provide acute care, and NICU environments are characterized by complexity, production pressure and need for co-ordination.

[a and b]

Thank you for this excellent advice, the first two paragraphs are now reduced to three sentences merged with the suggested paragraph that started on line 80. The intention is to provide a clearer focus on managers work. Figure 1 moved to line 114 for being closer to its first reference. Additionally, the third paragraph now clarify the “fluidity and structure balancing work” that managers do.

[c] Line 97-99 talks about how managers learn ... while this may be important it seems tangential to the goal of a study “..to explore how system resilience is enhanced by naturally occurring co-ordination..” Perhaps this section could be edited out.

[c] Removed

Methods: no new comments

Results: I think there are opportunities to improve the flow and internal logic of the results.

[2] Context: The results begin abruptly. In a revision they might begin with the time frame and some more explicit description of the participants, and the number approached (to give a sense of the representativeness of the sample studied) who was studied, and the volumes of data collected ... I found the immediate ‘look somewhere else’ was disruptive – as the authors directed me (the reader) to tables / other parts of the manuscript.

The first “introductory” paragraph in results have been expanded and revised for clarity. It now briefly mentions the data volumes. Rephrased and sorted the content for readability and more comprehensive introduction to the main- and sub-categories.

[3] Labels: For me the ‘manoeuvrability’ label does not fit well with what I think it describes: the situations that may ‘require’ adaption or perhaps (my preference) describe degrees of operational stress. I appreciate the conceptual model of structure and flexibility articulated as rationale for the study. From [i] usual comfortable ‘ok’ activity (baseline), to [ii] recognizing increasing stress on the available resources (problem recognition and standard responses) then to [iii] modifying usual practice / special adaptations/responses. I might be missing something about this label that others can see. Greater clarity about these ‘big categories’ –ideally presented early in the results- may help other readers and provide links back to the rationale for flexibility mentioned in the introduction.

Thank you for an excellent suggestion of a label for the second main category. It has been under debate for some time and we realise that the term “Manoeuvrability” might carry some interpretative power that we did not intend. We also removed “Losing control” from the subcategory “Losing control -Reorganisation” to avoid misinterpretations or negative connotations.

[4] Adjustments in Usual. Table 2: As I reflected on this important table I wondered about the rationale that 'adjustments' are needed for 'everyday work'. Perhaps this is a semantic / definitional issue, however to me it would seem that any adjustments would be from this 'usual' baseline. Reviewing the table I think the content of the cells in this column (the usual baseline situation) is either indirectly linked to resilience or might be described as the doing of mission-relevant activities. The authors may consider that a more inclusive description is that these columns represent activities rather than adjustments, and consider re-ordering the columns to reflect the transition from usual activities to special responses.

This is an important observation and greatly appreciated as we do not want to impose semantic problems with our nomenclature. The rationale for 'adjustments' are that adjustments are required to maintain ordinary conditions in a hazardous environment, however we realise the problem as you describe and have now renamed the sub-category to "Routine activities under ordinary conditions" to better reflect this aspect. We have also added a short description of the typical situation described under each level of operational stress on top the quote. This allowed us to remove table 3 (as suggested below in 'smaller points [a]').

[5] Core Mission: Interpretations about the (perhaps shared or divergent) understanding of 'core mission' (to inform operational priorities) by participants would strengthen the interpretation of the activities taken across the degrees of 'operational stress' from usual to special actions.

[a] I note in the discussion... 'agreed on making provision of acute care to rapidly deteriorating patients a top priority, allowing us to identify it as a core mission'. The statements about 'umbrella perspective' refer to "team members' ability to predict the most likely priorities of each other." Greater description of data/ observations about this 'ability' would strengthen the work.

The quote on line 186-190 are meant to reflect the acknowledgment of the inability to say no to patients that are born "in-house". Revised line 183 in results, now points toward this as a top priority. Line 330-335 of the mentioned paragraph are revised for clarity. Line 334-335 added.

[b] As I write this I wonder did the participants agree on these priorities (was this a shared mental model or an understanding of different mental models?). Were there observed moments of disagreement reflecting tensions between the interests/ priorities of the multiple teams mentioned.

[c] the statement : "we could not observe a formal agenda for how and why the management team was supposed to prioritize in terms of goal achievement below the core mission" may have implications for the 'formal' training of future managers, and suggests to me that individualized management /practice / co-ordination responses were common. Were participants asked about this more detailed prioritization? What did they say?

Indeed, the agreement between participants/stakeholders is a good point. Line 351-357 have been revised to provide a brief clarification on how we discussed this aspect.

[d] I think the measurement of successful decision-making / co-ordination comes back to the mission.... Was it achieved. The authors might reflect more on this if it aligns with the emergence of new data.

Paragraph starting on line 380 revised with this excellent suggestion in mind, we do want to express the problematic task of trying to define success in a complex (in-tractable) environment. Added a short reflection how coordinative work contributes to the system's capacity for expressing resilience.

[6] Conclusion: the first sentence seemed redundant, and other statements 'central to the capacity for expressing resilience' are (in my opinion) not well connected to the data. The final sentences about interventions and defining successful co-ordination are important, but diverge from the study findings. I suggest revising to reflect findings and gaps to be addressed in future work.

Conclusions are now revised to better align with the aim and build on the revised "discussion. Stronger focus on the main categories presented as two axes in table2.

[7] Discussion: might be expanded to better integrate understanding from other industries. Revisions of the "Supporting coherence" paragraph have been made with this comment in mind.

Line 325-329 added a short mention and reference to small-teams research mainly done in various industries.

Smaller points

[a] Table 3 might be added to table 2. **Done**

[b] The figure... perhaps decreasing the color intensity of the blue speech bubbles – I found they distracted from the main information at the center of the figure. **Done**